# Automatic detection of adult cardiomyocyte for high throughput measurements of calcium and contractility

L. Cao[1,2], E. Manders[1,3]*, M. Helmes[1,3]

**1** CytoCypher BV, Wageningen, The Netherlands, **2** Leiden Institute of Advanced Computer Science, University of Leiden, Leiden, The Netherlands, **3** Department of Physiology, VU University Medical Center, Amsterdam, The Netherlands

* emmy@cytocypher.com

**Data Availability Statement:** All relevant code are within the paper and its Supporting Information files. All cardiomyocyte measurement data are available on Kaggle www.kaggle.com/dataset/

## Abstract

Simultaneous calcium and contractility measurements on isolated adult cardiomyocytes have been the gold standard for the last decades to study cardiac (patho)physiology. However, the throughput of this system is low which limits the number of compounds that can be tested per animal. We developed instrumentation and software that can automatically find adult cardiomyocytes. Cells are detected based on the cell boundary using a Sobel-filter to find the edge information in the field of view. Separately, we detected motion by calculating the variance of intensity for each pixel in the frame through time. Additionally, it detects the best region for calcium and contractility measurements. A sensitivity of 0.66 ± 0.08 and a precision of 0.82 ± 0.03 was reached using our cell finding algorithm. The percentage of cells that were found and had good contractility measurements was 90 ± 10%. In addition, the average time between 2 cardiomyocyte calcium and contractility measurements decreased from 93.5 ± 80.2 to 15.6 ± 8.0 seconds using our software and microscope. This drastically increases throughput and provides a higher statistical reliability when performing adult cardiomyocyte functional experiments.

## Introduction

Isolated adult cardiomyocytes are widely accepted and extensively used as a model for cardiac physiology and pathophysiology [11]. Well-controlled in vitro environmental measurements can be performed on morphology, electrophysiology, biochemistry, gene regulation or contractile function [2]. These experiments are crucial in large drug or genetic screens that are related to cardiac (patho)physiology.

Several systems have been developed to measure the function of intact adult cardiomyocytes. Some systems are focussed on contractility [3–5], while others are able to measure calcium and contractility simultaneously [6, 7] or have a feedback mechanism to simulate cardiac work loops [4, 8]. These contractility and calcium handling measurements in vitro have been used as a platform for determining the physiological consequences of various genetic manipulations and identifying potential therapeutic targets for the treatment of heart failure [6]. It is a

dbcd572035084d44e770e84a6376b8c1e7496c9
90fd8af50e88c13de5f1978f8.

**Funding:** The funder provided support in the form of salaries for authors [LC, EM and MH], but did not have any additional role in the study design, data collection and analysis, decision to publish, or preparation of the manuscript. The specific roles of these authors are articulated in the 'author contributions' section.

**Competing interests:** CytoCypher BV has commercialized the algorithm described in this paper. This does not alter our adherence to PLOS ONE policies on sharing data and materials.

promising platform for cardiac research, but the full potential is inhibited by its low throughput.

Development of a high throughput system for high-content measurements of ratiometric calcium and contractility (sarcomere length) is challenging for 3 reasons. Firstly, measurements require a magnification of at least 20x, to have the optical resolution to visualize sarcomeres, but also a high numerical aperture, to capture sufficient fluorescence emission for calcium measurement. This precludes measuring tens or hundreds of cells per frame. Secondly, contractility measurements should be done at high speed. A contraction of a mouse cardiomyocyte paced at 2Hz takes <0.1 seconds [9, 10]. To reliably detect peak shortening and transient kinetics at least 250 frames per second (fps) are needed. Given the pixel clocks on standard cameras, this limits the camera chip size (in terms of megapixels), and thus the number of myocytes that can be measured simultaneously. Thirdly, each myocyte needs to be measured for at least 5–6 contractions to establish whether it is a regularly contracting myocyte. Depending on the pacing rate this takes minimally 2–6 seconds per cell. Therefore, the biggest gain in the number of cells that can be measured in an hour, is by reducing the time between measurements. With the existing systems it often takes minutes to find a new cell, to position it and adjust all necessary settings. As this is a manual process, selection of cells is subjective and may thus result in experimenter bias.

In this paper we show how you can dramatically increase throughput by developing efficient software, minimize travel time by moving the optics instead of the cell, and automating the cell finding and positioning process. With one click of a button the Multicell system, as shown in Fig 1, scans the well, identifies cardiomyocytes and measures calcium and contractility. Automated cell finding has the additional benefit that it standardizes cell selection and to a large extent removes experimenter bias. This dedicated system has the potential to dramatically improve cardiovascular research by providing an important tool for cardiomyocyte cellular physiology research and drug development.

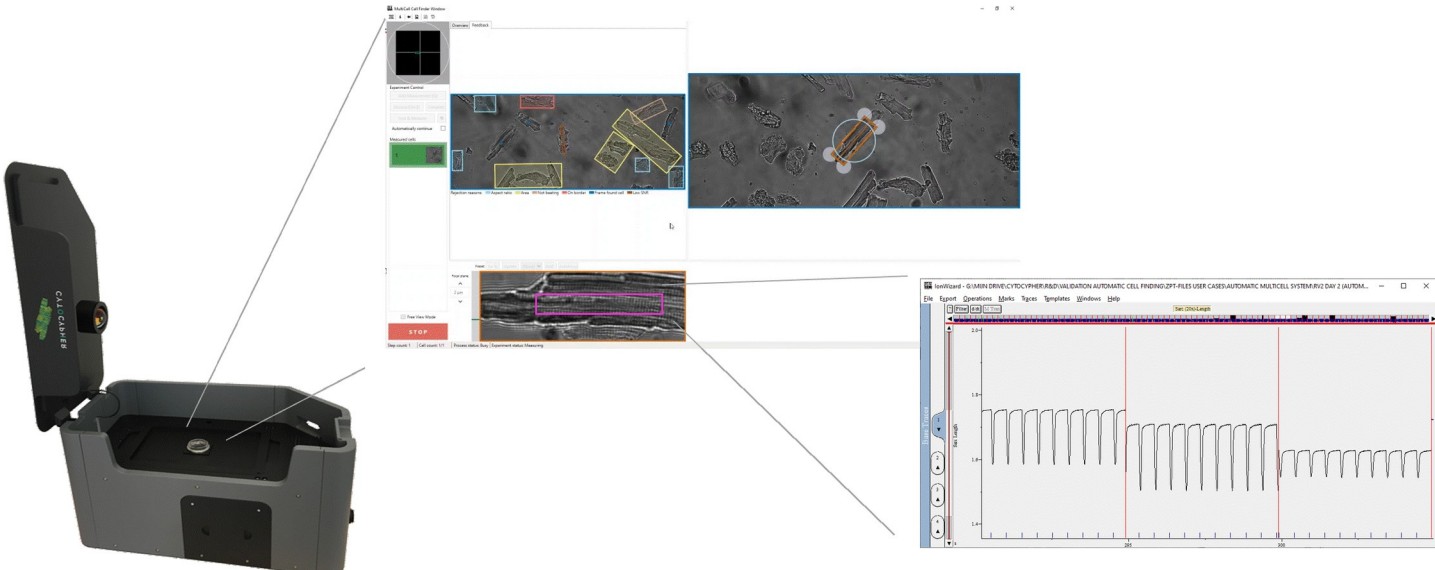

**Fig 1. Illustration of the MultiCell high throughput system.** A 35mm dish is placed in the microscope, MultiCell user interface showing the automatic cell finding feedback and the IonWizard acquisition window with sarcomere length traces.

## Methodology

### Experimental setup

**Adult rat ventricular cardiomyocyte isolation.** Animal experiments were performed in accordance with the guidelines from Directive 2010/63/EU of the European Parliament on the protection of animals used for scientific purposes and approved by the ethics committee of VU Medical Center, Amsterdam, the Netherlands.

Rat cardiomyocytes were isolated using Liberase digestion of hearts as described previously [11]. Briefly, adult male Wistar rats were anesthetized, the chest was opened and the heart was injected with cold EGTA solution. Then, the heart was quickly removed and perfused in a Langendorff set-up for 5 minutes. Next, it was perfused with enzyme solution until the heart was digested sufficiently. The right ventricle and atria were removed and the remaining part was cut into small pieces and titrated with a Pasteur pipet for 3 minutes. The cell suspension was filtered through a 300μm nylon mesh filter and re-suspended in $CaCl_2$ buffers of increasing calcium concentrations to reach a final concentration of 1mM. Isolated cardiomyocytes were resuspended in plating medium and seeded on laminin coated 35mm glass bottom dishes (MatTek Corporation, Ashland, MA, USA) and placed in an incubator for at least 30 minutes. Before measurements, the dish was washed with Tyrode to remove unattached cells.

**Cardiomyocyte calcium and contractility measurements.** Three different microscope setups for calcium and contractility measurements were used in this study. In all systems we used the MyoCam-S3, IonOptix software analysis 7.4.2 and a 20x objective (NA = 0.75). Cells were paced using MyoPacer (IonOptix LLC, Westwood, MA, USA), field stimulation settings were 2Hz, 4ms at 16V.

- Regular IonOptix Calcium and Contractility set-up (manual setup): Inverted microscope system where the user can move the dish with cells and adjust focus and cell orientation manually.

- CytoCypher MultiCell system (automatic setup): Inverted microscope system in which the objective can move in x-y-z position and reaches the next FOV within 100ms without having to move the dish with cells (Fig 1) [12, 13].

- A microscope equipped with a motorized stage (Prior Scientific Inc, Rockland, MA USA)). The same software is used as with the high speed microscope, but here the displacement speed is limited as the dish needs to be moved. The maximum acceleration without disturbing the cells is 3000 μm/s². The z-position of the objective is also controlled by the MultiCell software.

### Automatic cell finding

The pipeline we developed, using Emgu CV images processing library in C#, is specifically designed for a x, y and z-controllable objective or motorized stage microscope. Therefore, we are not able to share a working code in Github. Snippets of all the relevant code can be found in S1–S4 Files and are referred to in the sections below.

Fig 2 shows the overall structure of our automatic cell finding pipeline. It consists of 2 layers: Frame and Filter. The frame layer aims to find the cells in the whole field of view (FOV). We incorporated a static and a motion detection cell finding method. The filter layer consists of 3 filters to check the quality of the cell and to optimize the sarcomere length signal [14]. The objective or motorized stage moves to a new FOV once all cells have passed the filter layer. A spiral motion trajectory was chosen to move through the dish, with an overlap of 30% in both

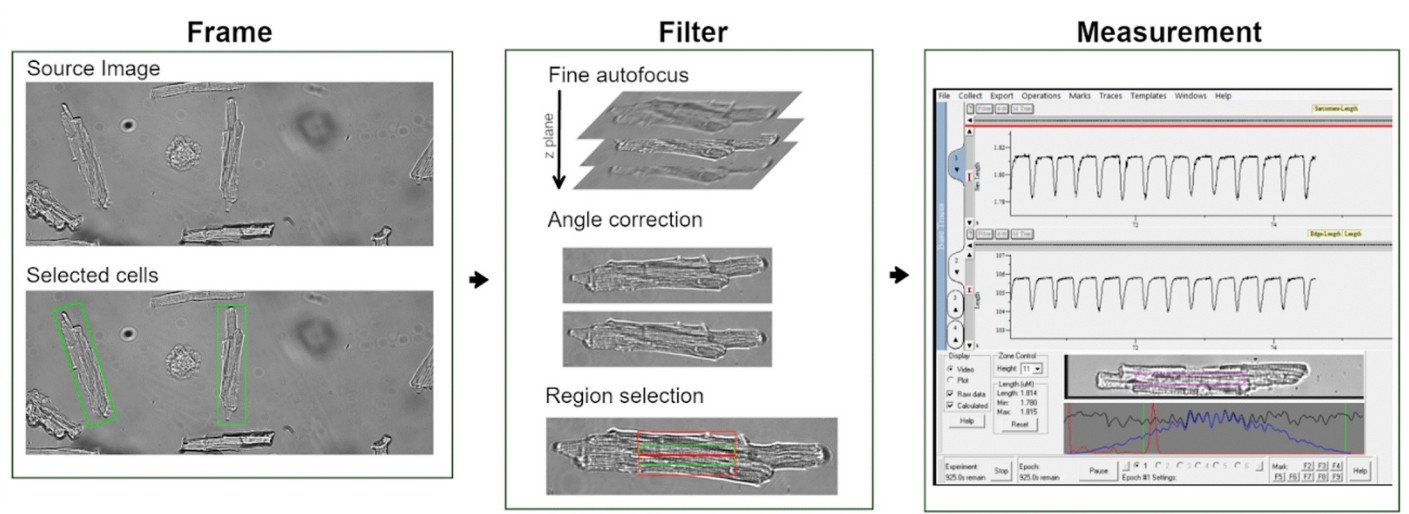

**Fig 2. Structure of automatic cell finding scheme.** Frame layer finds valid cells in the FOV. Filter layer finds the best subregion for sarcomere length measurement. It includes three main steps: fine autofocus, angle correction and region selection. Once the subregion is found, contractility measurement is conducted in this region, top trace shows sarcomere length, bottom trace shows cell length trace.

x- and y-direction to include cells that are on the border of the FOV. A more detailed description of the pipeline can be found below.

**Frame layer—static cell finding.** As shown in Fig 3, static cell finding detects cardiomyocytes based on the cell boundary irrespective of motion (S1 File). In a brightfield microscopy

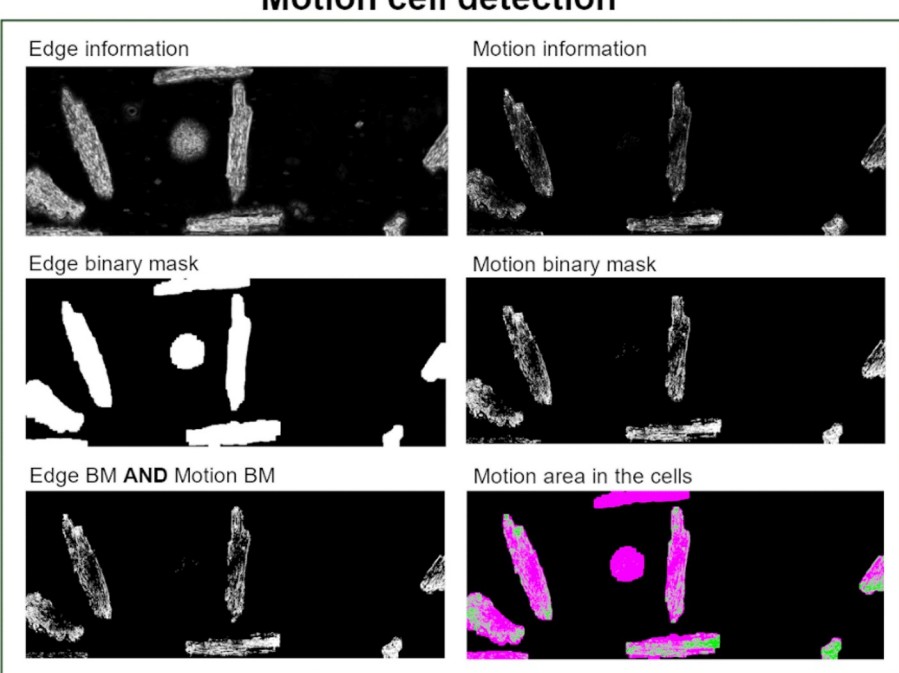

**Fig 3. Pipeline of static and motion cell finding.** Static cell detection only finds cells within a predefined minimum/maximum area and width and height ratio. Motion cell detection finds cells with sufficient motion information in it.

image, cardiomyocytes have strong edge information on the border of the cell as well as on the inside of the cell because of the sarcomere structure. Therefore, an edge filter is sufficient to identify a cell on a smooth background. In this application, we use a Sobel filter [15] to find the edge information. It follows up with a Gaussian filter (Radius: 5) and a contrast stretching filter (Saturated pixels: 0.4%) [16] in order to prepare for thresholding. The Otsu adaptive thresholding method [17] is chosen as our thresholding method for edge information. Subsequently the mathematical morphology operator "opening" is used on the binary mask to clean up the small objects and separate cells that are close to each other. The user can set the distance between two found cells i.e. for calcium measurements it is unwanted to have another cell located within the excitation spot.

**Motion cell finding.** One of the crucial requirements of our high throughput system is to limit the time between measurements. The minimum time required to detect motion is one beat, to which we want to stay as close as possible. This effectively reduces our computational time to approximately 1 second, which is challenging. Finding the motion of beating cells requires us to deal with a stream of frames instead of one frame. The most straightforward and intuitive way of finding cell motion is to calculate the variance of intensity for each pixel in the frame through time. Pixels with a high variance through time indicate a lot of motion. However, to calculate the variance you first need to know the mean of each pixel through time. Our system does not allow us to save the full frame rate (usually 250 fps) but only 20 fps for this calculation. The method would also not be fast enough for 250 fps. Therefore, we choose an efficient method to calculate the variance in an image stream called 'computations with shifted data' [18]. The argument is that the closer K is to the mean value the more accurate the result will be, but just choosing a value inside the samples range will guarantee the desired stability [19]. Following this logic, we treat the first frame as the mean for the variance calculation through time. From this we get the motion information out of the image stream (S2 File). The motion detection algorithm is described in a pseudocode as follows:

```
algorithm motion-detection is
input: number of frames n
output: image contains variation information through time variance
for i = 1 to n do
        get one frame f
        if i equals 1
            K = f
            initialize Ex as image with pixel intensity as zero
            initialize Ex2 as image with pixel intensity as zero
        else
            x = f
        Ex + = x—K
        Ex2 + = (x—K) * (x—K)
 variance = (Ex2 - (Ex * Ex) / n) / (n—1)
 normalize variance to range [0, 255]
    return variance
```

Only using motion information to find the beating cells is not sufficient because the beating signal in a cell is not homogeneously distributed, see Fig 3. Most of the time, the motion on the cell edge is larger than the motion in the cell center. A cell can sometimes get separated into multiple motion regions that fail to represent the whole beating cell region. That is why we first use edge information to find a cell and subsequently use motion information to indicate whether the cell is beating. In this second step we also calculate the ratio between motion and cell area for each cell, where we use a threshold of 5% to distinguish a beating cell from random pixel noise.

**Rejection criteria.** A list of rejection criteria is used to find valid cells. A valid cell should have a surface area between a user defined minimum and maximum, together with a user defined minimum and maximum aspect ratio of the bounding box. Furthermore, cells that are too close to the border of the FOV are rejected. The adjacency rejection criterium is that the four corners of the bounding box should not be within 10 pixels of the border. In addition, a ROI containing more than one cell is ruled out as well. As adult cardiomyocytes have a rectangular shape, we use the ratio of actual cell area divided by the area of the bounding box to define whether there is more than one cell included in the ROI. We set the threshold of the ratio to 0.55, any ROI that has a lower ratio than the threshold is rejected.

## Filter

Once the valid beating cells are found in the FOV, the microscope objective moves to center of one of the cells in the FOV. The cell is digitally reoriented in the FOV using a bilinear rotation algorithm [20] on the long edge of the bounding box for further fine tuning. The following filters are designed to adjust the position of the microscope objective so that the ideal subregion in the cell is centered in the FOV for sarcomere length measurement.

**Fine autofocus.** On the reoriented sub-image, we select a small region of interest in the center of the image (256x30) and average all the vertical lines in each column. Once we have a one-dimensional horizontal line, we perform a one dimensional Fast Fourier Transformation (1DFFT). The Fourier transformation technique is the same as IonOptix uses in their software [21]. In the frequency domain $f_i$, we select a range corresponding to a sarcomere length of 1–2μm. The peak ($Fmax_k$) and signal to noise ratio $FSNR_k$ is calculated in this range [$i_{min}$, $i_{max}$], see Eqs 1 to 4. The baseline $Fbase_k$ is defined as the higher value of the minima on the left- and right-hand side of the peak.

$$I_{BP} = \{I_k \ where \ MAX \ (Fmax_k)\} \tag{1}$$

$$Fmax_k = MAX \ (f_{i,k}), \ i \in [i_{min}, \ i_{max}] \tag{2}$$

$$I_{BP} = \{I_k \ where \ MAX(FSNR_k)\} \tag{3}$$

$$FSNR_k = \frac{Fmax_k}{Fbase_k} \tag{4}$$

To find the optimal focus plane for sarcomere structure, a fine autofocus filter with two sweeps is designed. The first sweep identifies the z-plane with the highest maximum peak value during a customized range in the z-direction, see Eqs 1 and 2. The second sweep identifies the z-plane with the best SNR in the frequency range, see Eqs 3 and 4. The range in the z-direction is smaller compared to the first sweep (S3 File).

**Angle correction.** A single cardiomyocyte is cropped by a rectangular region of interest (256x30). A 2DFFT is conducted and the parallel bands representing the sarcomere structure are visible [22]. Interestingly, if a second 2DFFT is conducted on this frequency image, we get a bright line indicating the orientation of the sarcomere structure in the cell [23]. We use this signal to find the ideal angle for the angle correction (S4 File).

**Region selection.** Three regions are tested for a maximum sarcomere signal (Fig 2: Region selection). The three candidate regions are one in the center, one 30 pixels above and one 30 pixels below the center region. A vertical line averaging is performed for each region upon which a 1DFFT is applied. The region with the highest SNR (Eq 4) is selected as the best region for sarcomere length measurement.

### Validation design

Two validation strategies are used to estimate the performance of our high throughput system. First, we validate whether the automatic cell finding pipeline can find the appropriate beating cells. Therefore, videos are recorded from 4 dishes of beating cells. What were considered appropriate beating cells were annotated manually by an expert. The manual annotations are compared to the automatic pipeline. Thereafter, sensitivity, precision and F1 score (see Eqs 5–7) are calculated to show the performance of automatic cell finding method.

$$Sensitivity = True\ Positives/(True\ Positives + False\ Negatives) \tag{5}$$

$$Precision = True\ Positives/(True\ Positives + False\ Positives) \tag{6}$$

$$F1\ score = 2 * (Sensitivity * Precision)/(Sensitivity + Precision) \tag{7}$$

Second, we compare the time it takes to measure cardiomyocytes using the original manual system versus the new automated microscope and algorithm.

**Case study.** With the MultiCell system we performed a case study using the optimal cell finding configuration to perform repeated measures on 20 cells before and after adding iso-prenaline (30nM). The experiment consists of 2 steps. First, we find and measure 20 cells automatically, next we add isoprenaline to the dish by manual pipetting (final concentration of 30nM), we incubate for 5 minutes, then the same cells are measured again. In each step fine autofocus is performed on the cell before measurement. Data are analysed using CytoSolver software (IonOptix LLC, United States).

## Result

### Validation automatic cell finding detection

In total we acquired 190 videos (total area of ~14mm2) of beating rat cardiomyocytes from 4 different dishes on 2 experimental days. Average cell density in these dishes was 65/mm2. In these 190 videos the automatic cell finding algorithm found 278 beating cells based on the pre-defined cell size (1500–5500 $\mu m^2$), length/width ratio (0.1–0.7) and motion. Of these 278 cells 66 were excluded based on a low SNR which was set at 2.5.

Each video was manually annotated, an example of 1 dish is shown in Fig 4. We calculated the Precision, Recall, F1 score based on the automatic and manually annotation of all dishes, see Table 1. It can be observed that precision increases considerably after inclusion of the SNR

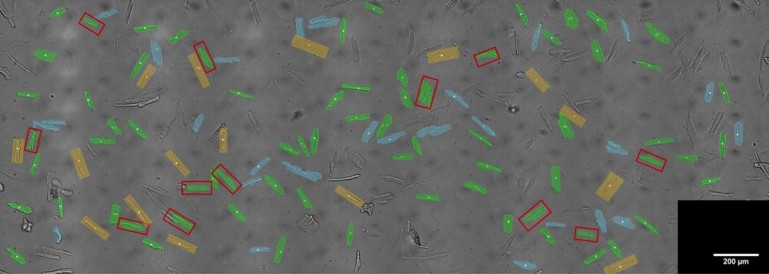

**Fig 4. Validation result of 1 dish of rat cardiomyocytes.** Image is a stitched and overlay view of 54 videos annotated manually (white dots) and found automatically by the algorithm (blue and green overlay i.e. True Positives). Blue overlay indicates cells which were filtered out for measurement based on a low SNR of the FFT signal. Red box indicates cells which were found by the software, but not annotated manually i.e. False Positives. Yellow boxes indicate cells that were annotated manually but not found by the algorithm i.e. False Negatives.

**Table 1. Validation results before and after inclusion of the signal to noise ratio filter.**

| | Dish 1 | Dish 2 | Dish 3 | Dish 4 | Average ± std |
|---|---|---|---|---|---|
| **Sensitivity** | 0.81 | 0.62 | 0.64 | 0.68 | 0.69 ± 0.09 |
| **Precision** | 0.72 | 0.64 | 0.74 | 0.71 | 0.70 ± 0.04 |
| **F1** | 0.77 | 0.63 | 0.69 | 0.70 | 0.69 ± 0.06 |
| Including signal to noise ratio filter | | | | | |
| | Dish 1 | Dish 2 | Dish 3 | Dish 4 | Average ± std |
| **Sensitivity** | 0.79 | 0.61 | 0.62 | 0.65 | 0.66 ± 0.08 |
| **Precision** | 0.83 | 0.79 | 0.81 | 0.87 | 0.82 ± 0.03 |
| **F1** | 0.81 | 0.69 | 0.70 | 0.74 | 0.73 ± 0.05 |
| **% of successful measurements** | 97.1 | 85.7 | 78.9 | 100 | 90.4 ± 9.9 |

filter, which is an indicator of cell quality. The percentage of cells that were found and had good contractility measurements was 90 ± 10%.

## Improvements in speed

By designing a fast x,y,z—programmable objective and accompanied graphical interface we considerably reduced the time between 2 measurements. As indicated in Table 2, the time between measurements reduced from 93 ± 80 seconds to 19 ± 12 seconds by using this microscope. Using the automatic cell finding pipeline further reduced this time to 15 ± 8 seconds. If we measure each cell for 5 seconds this brings us to ~180 cells per hour. It should be noted that depending on cell quality and cell density, the time between measurements can be further reduced to ~8 seconds, resulting in ~300 cells measured per hour.

## Case study with ISO

We validated the functionality of automatic cell finding by performing a standard cardiomyocyte experiment in which we added isoprenaline (30nM). The precision of the x, y, z-stage motor in the microscope allows us to perform repeated measurements of the same cells which were automatically found. Moving to already found cells and performing fine autofocus takes ~5 seconds. Therefore, performing a repeated measure on 25 cells takes less than 15 minutes, excluding incubation time.

The average contractility and calcium trace of 21 cells before and after isoprenaline is shown in Fig 5. Isoprenaline resulted in increased fractional shortening and speed of contraction (Fig 5A) as well as increased calcium released (Fig 5B). From Fig 5C and 5D it can be observed that there is cell-to-cell variation in response to isoprenaline.

## Experimenter bias

To illustrate that experimenter bias affects data collection we compared manual versus automatically acquired data in the same dish. In Fig 6A it can be observed that in most dishes

**Table 2. Speed of different systems.**

| | # of cells | Average time between measurements (s) |
|---|---|---|
| **Manual regular Calcium & Contractility system** | 23 | 93.5 ± 80.2 |
| **Manual CytoCypher Microscope** | 340 | 19.3 ± 12.3 |
| **Automatic CytoCypher Microscope** | 648 | 15.6 ± 8.0 |
| **Automatic Motorized Stage** | 641 | 19.2 ± 4.5 |

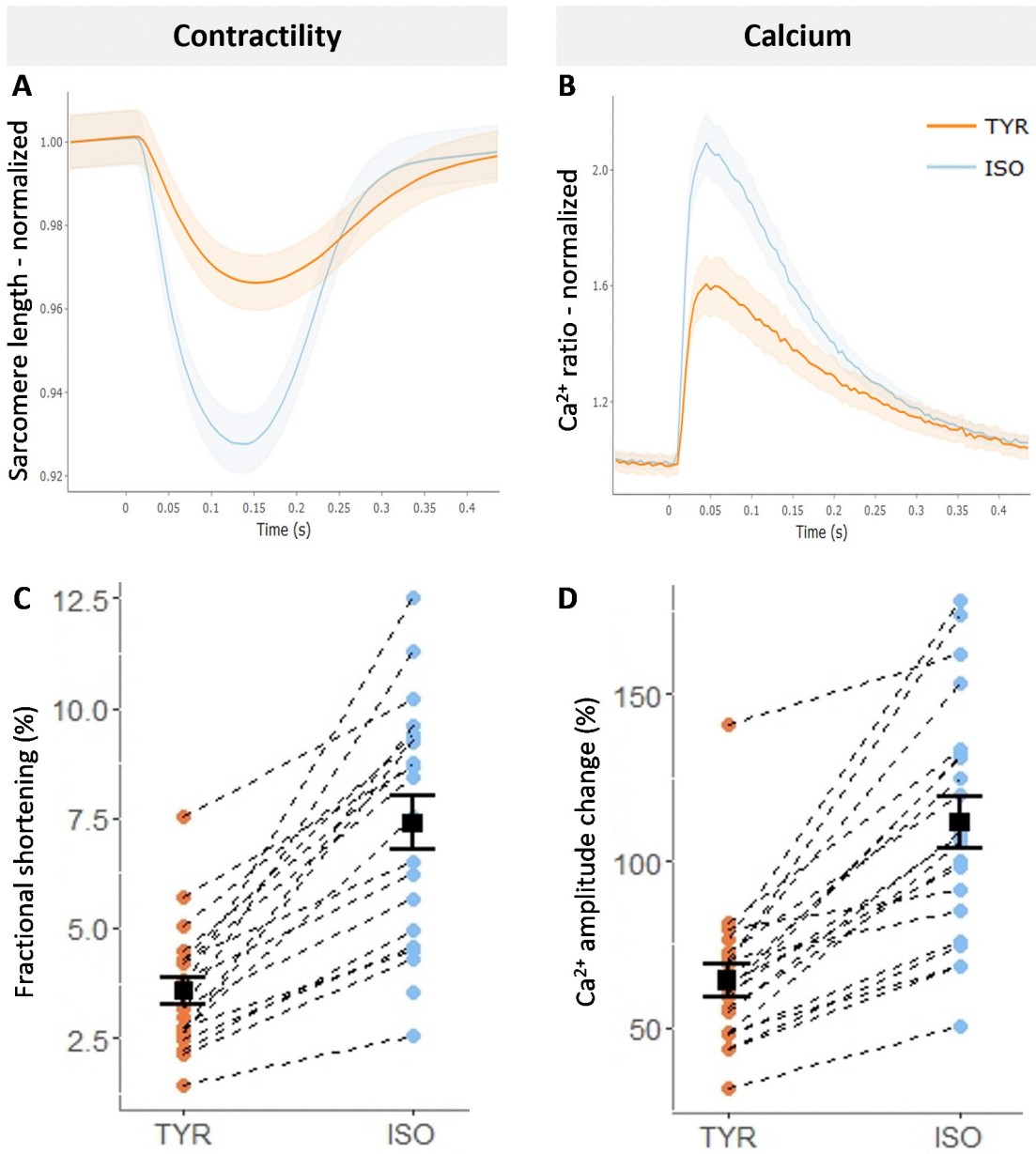

**Fig 5. Results isoprenaline (30nM) experiment. A**. Average contractility trace before and after isoprenaline (ISO, n = 21) shaded area indicates the 95% confidence interval. **B**. Average calcium trace before and after ISO (n = 21). **C.** Fractional shortening of each cell before and after ISO. **D.** Calcium amplitude change of each cell before and after ISO treatment.

baseline sarcomere length has a broader distribution with automatic cell detection compared to manually selected cells. In addition, fractional shortening distribution (Fig 6B) is shifted towards lower values in automatically found and measured cells compared to manually found cells.

## Discussion

We developed robust, highly automated software for detecting, moving towards, and measuring contraction and calcium transients in beating cardiomyocytes. This increased the

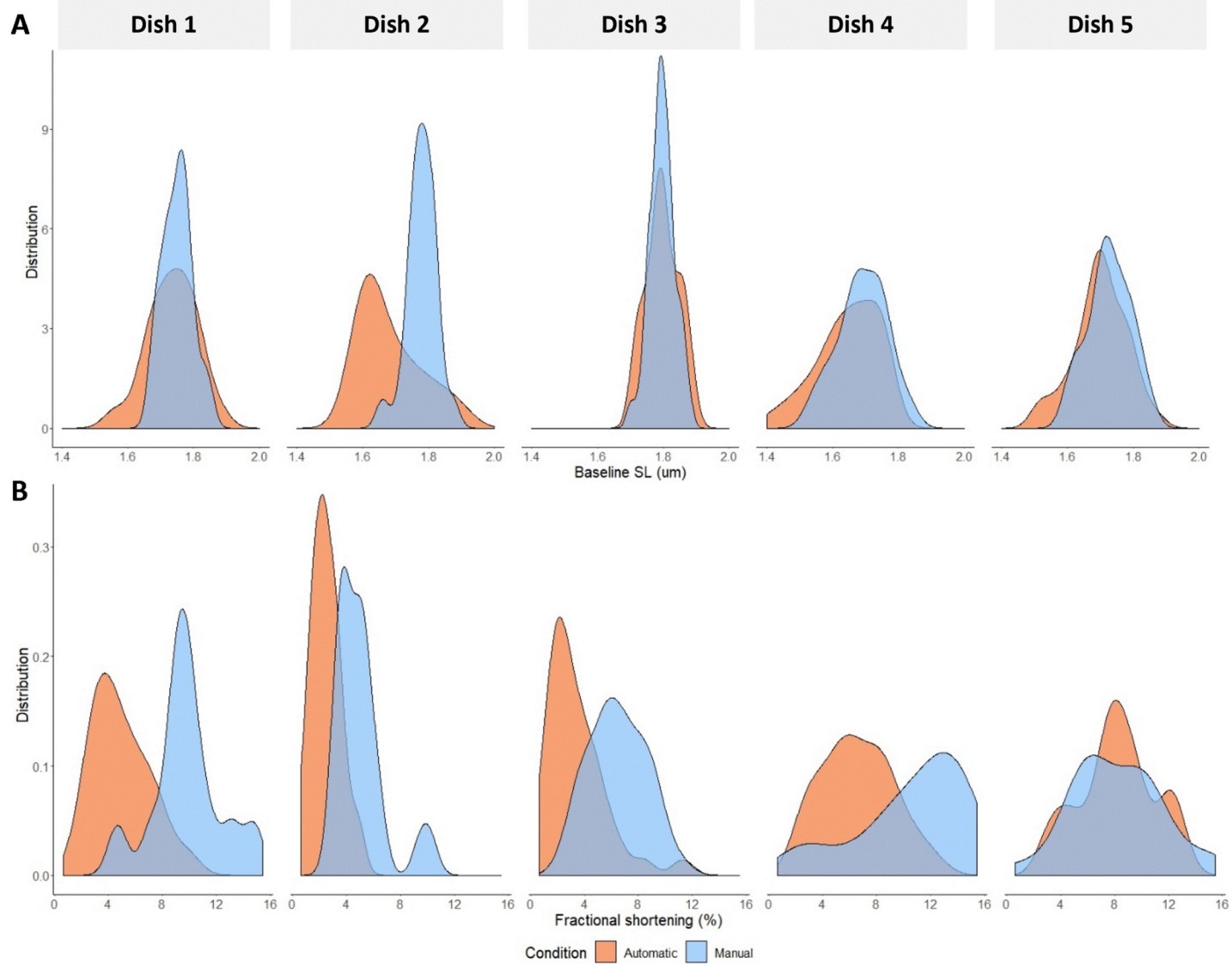

**Fig 6. Data distribution of 5 dishes measured manually and automatically. A.** Baseline sarcomere length (SL) distribution tends to shift to the left with automatically acquired data. **B.** Fractional shortening is shifted to the left in most dishes with automatically acquired data.

throughput of these measurements considerably. Wright et al. shows that this improves the data quality, enabling inter-preparation and within-preparation statistical correction which results in more reliable hypothesis testing [13, 24].

Cardiomyocyte cell isolations are used for new drug screens and to study cardiac (patho) physiology. From each rat or mouse millions of cells can be isolated, but in practice less than a hundred of these cells can be measured. Therefore, increasing the number of measurements per hour has large benefits. It would allow drug screens of 10–50 components performed on the same animal. In addition, the software stores the location of each cell which makes it easy to perform repeated measures on these cells under different conditions. The current validation was performed on rat cardiomyocytes, but Wright et al. also shows its potential in mouse, guinea pig and human cardiomyocytes [13]. The increased throughput reduces the number of

animals needed, the standardized conditions reduce variability between animals, and the increased number of measurements per drug result in more reliable outcomes [12, 13, 24].

Validation experiments showed that our algorithm was accurate and robust in multiple dishes with beating cardiomyocytes. In several dishes we manually annotated beating cardiomyocytes and compared this with automatically found cells. We scored well on precision (0.82 ± 0.03), but less on sensitivity (0.66 ± 0.08), see Table 1. A low sensitivity is unlikely to lead to a bias in the acquired data as the number of cells measured is predefined. However, it does affect the speed of the automatic pipeline. If cells are skipped, a larger area of the dish needs to be scanned to measure the specified number of cells. We managed to increase precision by 0.12 when we added a SNR filter to assess the quality of the cells. Precision can be further increased by narrowing the inclusion criteria of cell size, width/height ratio and increasing SNR. However, this will also reduce sensitivity and one needs to balance the two in order to have a reasonable amount of cells within 1 hour. This balance can change between projects, as it depends on the quality of the cells. Additionally, the acquired data can be filtered by applying exclusion criteria on for example the number of transients, baseline sarcomere length or goodness of transient fitting in CytoSolver [24].

From field studies, where each cell was measured once, most time was saved (a reduction of ~75 seconds between subsequent cells) by having a motorized microscope with a user-friendly software interface. This enables you to control the microscope and perform digital image rotation, thus no need to switch back and forth between computer and microscope. The second biggest factor was the design where the microscope objective moves rather than the dish with cells, which reduced the time between measurements by another ~4 seconds. By automating the cell finding process an additional time saving of ~4 seconds was achieved. Although the time gain of automating the cell finding process compared to manually selecting the cells is limited, automation has the advantages of removing the experimenter bias and freeing up time for the researcher to perform other tasks simultaneously.

One of the current limitations is that cells on the border of a frame are excluded, therefore an overlap of at least 30% on each frame is necessary to capture these cells. This limits the area scanned per frame which is already limited by the large magnification needed to capture the sarcomere structure. Another limitation is that it is difficult to get an even spread of the cells throughout a dish. The pipeline must pause for at least 1 second to detect motion for each frame. Each frame that only contains overlapping cells or no viable cells is wasted time. A possible solution is to create a stitched image of multiple FOV and apply object detection on this image. Next, each cell is checked for motion and while measuring this cell the next cell in the same FOV could be assessed for motion. This approach has been modelled and our calculations suggest that this should reduce the time between measurements to ~6 seconds. Another approach is to ensure equal cell distribution by plating the cells on a regular laminin grid or trapping them inside a polydimethylsiloxane microwell [25]. This could result in 6 cells within one FOV, which would reduce the time between measurements to ~5.2 seconds.

Furthermore, the current static cell finding algorithm is a rule-based method and mostly rely on the information of edge. The static cell finding algorithm can be further improved by using deep learning models [26, 27] where more features are incorporated to identify the cell in the FOV. These models show superior performance in nuclei detection in 2018 Data Science Bowl. In addition, cardiac specialists could be involved in annotating the healthy cardiomyocytes from the images as training data. Some pre-liminary experiments (not shown here) show very promising results using deep learning models for adult cardiac myocytes.

Determining the optimal focal plane on the sarcomere structure takes ~5 seconds as we need to perform a complete sweep in a large z range while calculating the SNR of the FFT signal. Currently, we perform these calculations on 20fps instead of the full 250fps. This limits the

speed of moving the objective stage. When it is possible to perform these calculations on 250fps it would only take ~0.5 seconds and would reduce the time between measurements to 2–3 seconds. Resulting in 450–500 cells per hour when you would measure a cell for 5 seconds.

The advantage of a motorized microscope is that the position of the cell is saved allowing repeated measurements on multiple cells, as illustrated in our case study with isoprenaline (Fig 5). Instead of following 1 cell over the time course of 15 minutes, you can now follow as many as the stability of the cells allows. The advantage of repeated measurements is that it enables you to detect responders from non-responders which could be vital information during drug development. In addition it gives more statistical power as it controls for factors that cause variability between animals and cells [28, 29].

Blind testing of compounds or with different genetic models is in practice hard in adult cardiomyocytes. Automatically selecting and measuring cells helps to remove experimenter bias and thus improve data quality. From our data set we can see that data distribution is different between manually and automatically acquired data. This was expected as you manually tend to select the strongest beating cells and discard cells that have a small baseline sarcomere length. Partly, this is justified as a short baseline sarcomere length typically indicates damaged cells. However, small contractions might be part of the genetic model or the effect of a drug.

In conclusion, the MultiCell high throughput microscope and software for adult cardiomyocytes allows us to automatically find and measure hundreds of cells per hour. This is valuable as it improves data quality, reduces the number of animals per experiment and frees up valuable time of the researcher.

## Supporting information

**S1 File. Static cell detection.**
(PDF)

**S2 File. Motion cell detection.**
(PDF)

**S3 File. Fine auto focus.**
(PDF)

**S4 File. Angle correction.**
(PDF)

## Acknowledgments

The authors acknowledge Menne van Willigenburg, MSc, Bob van Hoek MSc and Max Goebel BSc for the provided technical support.

## Author Contributions

**Conceptualization:** L. Cao, E. Manders, M. Helmes.

**Data curation:** E. Manders.

**Formal analysis:** L. Cao, E. Manders.

**Investigation:** L. Cao.

**Methodology:** L. Cao, E. Manders.

**Software:** L. Cao, E. Manders.

**Supervision:** M. Helmes.

**Validation:** E. Manders.

**Visualization:** E. Manders.

**Writing – original draft:** L. Cao, E. Manders.

**Writing – review & editing:** L. Cao, M. Helmes.

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
