## [Decision Letter · Decision Letter 0]

12 Jul 2021

PONE-D-21-19111

Automatic detection of adult cardiomyocyte for high throughput measurements of calcium and contractility

PLOS ONE

Dear Dr. Manders,

Thank you for submitting your manuscript to PLOS ONE. After careful consideration, we feel that it has merit but does not fully meet PLOS ONE’s publication criteria as it currently stands. Therefore, we invite you to submit a revised version of the manuscript that addresses the points raised during the review process.

If you could address the minor corrections requested by the reviewers, we would be happy to publish your manuscript.

We look forward to receiving your revised manuscript.

Kind regards,

Jean-Christophe Nebel, Ph.D

Academic Editor

PLOS ONE

Journal Requirements:

2. In the competing interests statement, please clarify why your affiliations to CytoCypher BV represent a competing interests.

"I have read the journal's policy and the authors of this manuscript have the following competing interests:L. Cao was an employee of CytoCypher BV, E. Manders is a part-time employee of CytoCypher BV and M.Helmes is the CEO of CytoCypher BV"

We note that one or more of the authors are employed by a commercial company: CytoCypher BV.

3.1. Please provide an amended Funding Statement declaring this commercial affiliation, as well as a statement regarding the Role of Funders in your study. If the funding organization did not play a role in the study design, data collection and analysis, decision to publish, or preparation of the manuscript and only provided financial support in the form of authors' salaries and/or research materials, please review your statements relating to the author contributions, and ensure you have specifically and accurately indicated the role(s) that these authors had in your study. You can update author roles in the Author Contributions section of the online submission form.

3.2. Please also provide an updated Competing Interests Statement declaring this commercial affiliation along with any other relevant declarations relating to employment, consultancy, patents, products in development, or marketed products, etc.  

Reviewers' comments:

Reviewer's Responses to Questions

**Comments to the Author**

1. Is the manuscript technically sound, and do the data support the conclusions?

Reviewer #1: Yes

Reviewer #2: Yes

2. Has the statistical analysis been performed appropriately and rigorously? 

Reviewer #1: Yes

Reviewer #2: Yes

3. Have the authors made all data underlying the findings in their manuscript fully available?

Reviewer #1: No

Reviewer #2: Yes

4. Is the manuscript presented in an intelligible fashion and written in standard English?

Reviewer #1: Yes

Reviewer #2: Yes

5. Review Comments to the Author

Reviewer #1: 1)

The model is demonstrated in a scenario with a low cell density (Fig. 3). I am wondering how the performance of the automatic segmentation masks would change in a scenario with high cell density. As the algorithm is based on edge detection, one might expect a drastic drop in segmentation performance (F1 score) in a scenario with overlapping cells / cell agglomerations where e.g. watershed seg. / deep learning models are needed.

In Line 335 you are stating „Each frame that only contains overlapping cells or no viable cells is wasted time“.

Regarding that I have some questions:

How important are scenarios of high cell density (overlapping cells) for the measurements? Are only regions with low cell densities (no overlapping cells) of interest for the measurement? Is a measurement with a cell confluency of 100% / high cell density possible?

Is there a post-processing step needed (manual or automated) to exclude data / frames containing these "unnecessary data" (overlapping cells / agglomerations / image artifacts)? In other words, what is the process for image data selection?

2)

In the Result section you mention that the total amount of data acquired were 190 videos. Out of these videos, 212 beating cells are obtained / measured (line 232 ff.). Do you have an estimation / do you know the number of total cells contained in these 190 videos? In other words, what is the rate of total cells to actually measured cells? Can you upload some sample videos / images?

Reviewer #2: In this manuscript, Cao et al describe an image analysis workflow capable of automatically identifying viable isolated adult cardiomyocytes and extracting sarcomere shortening and Ca data from each identified cell. In conjunction with computer-controlled microscopy equipment, this image analysis approach enables a substantial increase in throughput (cells per unit time). It also removes experimental bias by using consistent and quantitative metrics for quality.

The work presented here is certainly of interest to the field and communicates some interesting and novel approaches to detecting and characterizing isolated cardiomyocytes. I have a few minor comments that I feel should be addressed:

1. Lines 280-288: In describing data histograms, the authors speak of distributions being “shifted to the left” – this is a qualitative description that doesn’t mean anything to the reader. It would be much better to say something like “the distribution of fractional shortening measurements was shifted toward smaller values of shortening in the automatically found cells…”

2. Discussion, line 292: The authors mention that their approach “improved the data quality and resulted in more reliable hypothesis testing.” I would encourage the authors to be much more specific or to remove this language. It is not obvious what “quality” means, nor what hypotheses were being tested – I don’t see where the data prove that the automated testing resulted in a more reliable hypothesis test.

3. Discussion, lines 314-318: There are obviously various parameters in the image analysis that can be varied in order to optimize output. Are the values reported and used in this manuscript likely to work generally across projects, or will this have to be revisited for cells from different species, disease states, etc? It would be helpful to comment on this.

4. Our own work has involved improving throughput of these isolated cardiomyocyte measurements. I noticed that exchange of solutions to alter drug presence was done manually by hand. We developed a microwell system for trapping cardiomyocytes in predictable orientations while also enabling continuous fluid flow (and exchange). Please see https://doi.org/10.1016/j.bpj.2019.08.024. It may be helpful to consider how this approach could be used to extend the methods presented here.

6. PLOS authors have the option to publish the peer review history of their article (what does this mean?). If published, this will include your full peer review and any attached files.

Reviewer #1: No

Reviewer #2: **Yes: **Stuart G. Campbell

---

## [Author Response · Author response to Decision Letter 0]

26 Jul 2021

Response to reviewers of the manuscript titled “Automatic detection of adult cardiomyocyte for high throughput measurements of calcium and contractility.”

We thank the reviewers and the editor for their careful consideration of our work and appreciate the criticisms and recommendations, which we believe have helped to improve the manuscript.

R1- We have carefully checked the style requirements and have made some small modifications to the file naming and reference list. 

2. In the competing interests statement, please clarify why your affiliations to CytoCypher BV represent a competing interests.

R2 - CytoCypher BV has commercialized the algorithm described in this manuscript producing cell finding software.

"I have read the journal's policy and the authors of this manuscript have the following competing interests:L. Cao was an employee of CytoCypher BV, E. Manders is a part-time employee of CytoCypher BV and M.Helmes is the CEO of CytoCypher BV"

We note that one or more of the authors are employed by a commercial company: CytoCypher BV.

3.1. Please provide an amended Funding Statement declaring this commercial affiliation, as well as a statement regarding the Role of Funders in your study. If the funding organization did not play a role in the study design, data collection and analysis, decision to publish, or preparation of the manuscript and only provided financial support in the form of authors' salaries and/or research materials, please review your statements relating to the author contributions, and ensure you have specifically and accurately indicated the role(s) that these authors had in your study. You can update author roles in the Author Contributions section of the online submission form.

3.2. Please also provide an updated Competing Interests Statement declaring this commercial affiliation along with any other relevant declarations relating to employment, consultancy, patents, products in development, or marketed products, etc. 

R3 - We will add the Funding Statement to our submission. We have updated our Competing Interests Statement as suggested. 

 

Reviewer #1: 

1 a) The model is demonstrated in a scenario with a low cell density (Fig. 3). I am wondering how the performance of the automatic segmentation masks would change in a scenario with high cell density. As the algorithm is based on edge detection, one might expect a drastic drop in segmentation performance (F1 score) in a scenario with overlapping cells / cell agglomerations where e.g. watershed seg. / deep learning models are needed.

R1a - The algorithm is specifically designed for measurements of calcium and contractility of single cardiomyocytes. These measurements require the cells not to overlap or touch each other as that could influence these measurements. We have added the average cell density of the dishes used for validation of our method (see line 235). In the discussion we explain in more detail how differences in cell density could affect the precision and sensitivity of our algorithm. See lines 337-346.

1 b) In Line 335 you are stating „Each frame that only contains overlapping cells or no viable cells is wasted time“.

Regarding that I have some questions:

How important are scenarios of high cell density (overlapping cells) for the measurements? Are only regions with low cell densities (no overlapping cells) of interest for the measurement? Is a measurement with a cell confluency of 100% / high cell density possible?

R1b - Indeed, only regions with no overlapping cells are of interest for these measurements. These are isolated cardiomyocytes which are not kept in culture but are measured on day of isolation. It is possible to measure calcium and contractility in a confluent cell layer, for example from cardiomyocyte induced pluripotent stem cells. However, the method of finding regions of interest to measure in these constructs is different and should not be based on edge detection.

1 c) Is there a post-processing step needed (manual or automated) to exclude data / frames containing these "unnecessary data" (overlapping cells / agglomerations / image artifacts)? In other words, what is the process for image data selection?

R1c - The current cell finding algorithm described in this paper includes a filter for overlapping cells (see lines 172-176) and enables the user to set a threshold for the distance between cells. 

We thank the reviewer for pointing this out. We have clarified this more in the revised manuscript on lines 135-137.

2) In the Result section you mention that the total amount of data acquired were 190 videos. Out of these videos, 212 beating cells are obtained / measured (line 232 ff.). Do you have an estimation / do you know the number of total cells contained in these 190 videos? In other words, what is the rate of total cells to actually measured cells? Can you upload some sample videos / images?

R2 - We thank the reviewer for pointing this out. We added the cell density of the dishes used (62, 58, 70, 73 = 65/mm2). Although, it should be noted that for these measurements we are only interested in non-overlapping contracting cells and the distribution can vary a lot within a dish; cells tend to cluster together in the middle of dish. In our opinion, the number of total cells within a dish is not that relevant to assess the performance of our algorithm. However, we do feel it is valuable information to the user to report the cell density see line 235. In addition, we added the videos for the validation to the data repository. 

Reviewer #2: 

In this manuscript, Cao et al describe an image analysis workflow capable of automatically identifying viable isolated adult cardiomyocytes and extracting sarcomere shortening and Ca data from each identified cell. In conjunction with computer-controlled microscopy equipment, this image analysis approach enables a substantial increase in throughput (cells per unit time). It also removes experimental bias by using consistent and quantitative metrics for quality.

The work presented here is certainly of interest to the field and communicates some interesting and novel approaches to detecting and characterizing isolated cardiomyocytes. I have a few minor comments that I feel should be addressed:

We thank the reviewer for his appreciation of our work.

1. Lines 280-288: In describing data histograms, the authors speak of distributions being “shifted to the left” – this is a qualitative description that doesn’t mean anything to the reader. It would be much better to say something like “the distribution of fractional shortening measurements was shifted toward smaller values of shortening in the automatically found cells…”

R1 - We agree with the reviewer that we could give a better description of the changes in the data histograms. We have adjusted this paragraph accordingly (Results, lines 285-286) 

2. Discussion, line 292: The authors mention that their approach “improved the data quality and resulted in more reliable hypothesis testing.” I would encourage the authors to be much more specific or to remove this language. It is not obvious what “quality” means, nor what hypotheses were being tested – I don’t see where the data prove that the automated testing resulted in a more reliable hypothesis test.

R2 - We thank the reviewer for pointing this out, we agree that our wording was not specific enough. We have adjusted this paragraph accordingly (Discussion, lines 295-297)

3. Discussion, lines 314-318: There are obviously various parameters in the image analysis that can be varied in order to optimize output. Are the values reported and used in this manuscript likely to work generally across projects, or will this have to be revisited for cells from different species, disease states, etc? It would be helpful to comment on this.

R3 - You can set the various parameters in way they can be used across projects. The settings used in the manuscript functioned well for both mouse and rat isolations. For isolations from different species (i.e. human, porcine), you may require to revisit the settings. The settings can be saved and can easily be imported when starting an experiment to ensure the same settings are used within a project. 

4. Our own work has involved improving throughput of these isolated cardiomyocyte measurements. I noticed that exchange of solutions to alter drug presence was done manually by hand. We developed a microwell system for trapping cardiomyocytes in predictable orientations while also enabling continuous fluid flow (and exchange). Please see https://doi.org/10.1016/j.bpj.2019.08.024. It may be helpful to consider how this approach could be used to extend the methods presented here.

R4 - Indeed, your work on trapping isolated cardiomyocytes could further improve the throughput of our algorithm. It ensures equal distribution of the cells throughout a dish and prevents or reduces the chances of cells overlapping. We address this in the discussion on lines 343-346.

---

## [Decision Letter · Decision Letter 1]

13 Aug 2021

Automatic detection of adult cardiomyocyte for high throughput measurements of calcium and contractility

PONE-D-21-19111R1

Dear Dr. Manders,

We’re pleased to inform you that your manuscript has been judged scientifically suitable for publication and will be formally accepted for publication once it meets all outstanding technical requirements.

Kind regards,

Jean-Christophe Nebel, Ph.D

Academic Editor

PLOS ONE

Additional Editor Comments (optional):

Reviewers' comments:

Reviewer's Responses to Questions

**Comments to the Author**

1. If the authors have adequately addressed your comments raised in a previous round of review and you feel that this manuscript is now acceptable for publication, you may indicate that here to bypass the “Comments to the Author” section, enter your conflict of interest statement in the “Confidential to Editor” section, and submit your "Accept" recommendation.

Reviewer #1: All comments have been addressed

Reviewer #2: All comments have been addressed

2. Is the manuscript technically sound, and do the data support the conclusions?

Reviewer #1: Yes

Reviewer #2: Yes

3. Has the statistical analysis been performed appropriately and rigorously? 

Reviewer #1: Yes

Reviewer #2: Yes

4. Have the authors made all data underlying the findings in their manuscript fully available?

Reviewer #1: Yes

Reviewer #2: Yes

5. Is the manuscript presented in an intelligible fashion and written in standard English?

Reviewer #1: Yes

Reviewer #2: Yes

6. Review Comments to the Author

Reviewer #1: All of the comments regarding Revision Number 1 have been adressed satisfyingly.

Reviewer #2: (No Response)

7. PLOS authors have the option to publish the peer review history of their article (what does this mean?). If published, this will include your full peer review and any attached files.

Reviewer #1: **Yes: **Fabian Englbrecht

Reviewer #2: **Yes: **Stuart Campbell

---

## [Editor Report · Acceptance letter]

23 Aug 2021

PONE-D-21-19111R1 

Automatic detection of adult cardiomyocyte for high throughput measurements of calcium and contractility 

Dear Dr. Manders:

I'm pleased to inform you that your manuscript has been deemed suitable for publication in PLOS ONE. Congratulations! Your manuscript is now with our production department. 

Kind regards, 

on behalf of

Prof. Jean-Christophe Nebel 

Academic Editor

PLOS ONE